# Conformer-Based Dental AI Patient Clinical Diagnosis Simulation Using Korean Synthetic Data Generator for Multiple Standardized Patient Scenarios

**DOI:** 10.3390/bioengineering10050615

**Published:** 2023-05-19

**Authors:** Kangmin Kim, Chanjun Chun, Seong-Yong Moon

**Affiliations:** 1Department of Computer Engineering, Chosun University, Gwangju 61452, Republic of Korea; could714@chosun.ac.kr (K.K.); cjchun@chosun.ac.kr (C.C.); 2Department of Oral and Maxillofacial Surgery, College of Dentistry, Chosun University, Gwangju 61452, Republic of Korea

**Keywords:** clinical medicine, artificial intelligence, dental informatics, conformer

## Abstract

The goal of clinical practice education is to develop the ability to apply theoretical knowledge in a clinical setting and to foster growth as a professional healthcare provider. One effective method of achieving this is through the utilization of Standardized Patients (SP) in education, which familiarizes students with real patient interviews and allows educators to assess their clinical performance skills. However, SP education faces challenges such as the cost of hiring actors and the shortage of professional educators to train them. In this paper, we aim to alleviate these issues by utilizing deep learning models to replace the actors. We employ the Conformer model for the implementation of the AI patient, and we develop a Korean SP scenario data generator to collect data for training responses to diagnostic questions. Our Korean SP scenario data generator is devised to generate SP scenarios based on the provided patient information, using pre-prepared questions and answers. In the AI patient training process, two types of data are employed: common data and personalized data. The common data are employed to develop natural general conversation skills, while personalized data, from the SP scenario, are utilized to learn specific clinical information relevant to a patient’s role. Based on these data, to evaluate the learning efficiency of the Conformer structure, a comparison was conducted with the Transformer using the BLEU score and WER as evaluation metrics. Experimental results showed that the Conformer-based model demonstrated a 3.92% and 6.74% improvement in BLEU and WER performance compared to the Transformer-based model, respectively. The dental AI patient for SP simulation presented in this paper has the potential to be applied to other medical and nursing fields, provided that additional data collection processes are conducted.

## 1. Introduction

Recent sociodemographic shifts, advances in science and technology, and heightened interest in health are rapidly transforming the healthcare environment globally. As the health management environment reaches a major turning point, the demand for medical personnel with exceptional job skills is increasing. Clinical practice education aims to cultivate various qualities for effectively functioning as competent medical personnel based on theoretical knowledge [1]. One of the practical education methods for this is simulation-based medical education. Generally, simulation-based education is actively utilized in the fields of medical and nursing education as it complements limited clinical training and provides a safe environment for practice in a setting similar to real-world clinical situations. Compared to traditional lecture-based education, the biggest advantage of simulation education is that it not only improves learners’ knowledge, but also effectively enhances problem-solving abilities and clinical performance skills [2].

Standardized patient (SP) is one of the simulation education methods. It utilizes actors trained to reproduce an actual patient’s medical history and physical examination results to educate and evaluate students’ clinical performance skills. SP is a reliable tool for effectively conducting interviews with real patients who have no substitute in student and resident education, and for assessing their clinical practice skills [3]. Due to these advantages, SP techniques are actively used in Objective Structured Clinical Examinations (OSCE) around the world. However, utilizing actors as SPs for practice comes with the requirement of expensive labor costs, dedicated manpower, and significant time for actor training [4]. As deep learning has advanced, there have been reports of research utilizing it for disease diagnosis [5], but applications for clinical diagnosis simulation are rare. In this study, we aim to utilize deep learning to alleviate the reliance on actors in the SP method.

Deep learning has become the core technology of the fourth industrial revolution thanks to advances in algorithms, increased computing power, and the use of large-scale datasets. Specifically, in the field of Natural Language Processing (NLP), which enables computers to understand human language, the introduction of deep learning has demonstrated its superior performance compared to existing rule-based and statistical methods.

Recurrent Neural Network (RNN) is a structure that carries information from the previous time step to the next, marking the beginning of neural-network-based continuous data processing. RNN has a long-term dependency problem, where vanishing gradients occur during backpropagation as the input sequence lengthens; Long Short-Term Memory (LSTM) and Gated Recurrent Unit (GRU), which were proposed later, introduced a memory cell, which mitigated the long-term dependency problem [6,7].

RNN-based systems are utilized for constructing encoders and decoders in Sequence-to-Sequence (Seq2Seq) models. These models are applied in various areas, including machine translation, speech-to-speech translation, and text summarization, where the goal is to transform an input sequence into a sequence in another domain [8,9,10]. However, Seq2Seq models have a limitation, as input information may be lost during the process of compressing the data into a fixed-size context vector, which is then passed to the decoder. To address this issue, attention mechanisms have been introduced [11,12].

In the attention mechanism, the decoder references the encoder’s information related to the predicted word each time it generates the current output word. This approach helps alleviate the issue of information loss by providing additional contextual information for each word. The attention mechanism’s introduction breathed new life into NLP and influenced model architecture. The Transformer [13] is an encoder–decoder structure that replaced the RNN-based architecture with an attention mechanism. This design allows for parallel processing and computation of word relationships, leading to improved machine translation performance and accelerated learning speed.

The remarkable architecture of the Transformer has applications beyond language-related fields, such as large-scale language models and Text-to-Speech (TTS), extending to image processing domains as well [14,15,16,17]. Recently, in the domain of audio signal processing, the Conformer model has been introduced [18], which is a combination of a Convolutional Neural Network (CNN) and Transformer. The Conformer, capable of capturing both local and global features of data, has been applied to speech recognition and speech separation, demonstrating superior performance compared to the Transformer and attracting significant research interest [18,19]. The Conformer’s comprehensive data feature capture abilities are anticipated to be effective in not only speech but also NLP. Accordingly, we developed an AI patient model that employs the Conformer structure.

The AI patient model learns from text-based conversational data, divided into common and personalized data that reflect the unique aspects of doctor–patient communication. Doctor–patient communication, focusing on the patient’s expression of symptoms and the doctor’s diagnostic actions, plays a critical role in the treatment process, as diagnosis and treatment outcomes may vary depending on the quality of communication [20]. The doctor must rely entirely on the patient’s description to evaluate their condition; however, the burden of expressing their symptoms and fear of the doctor may hinder smooth communication [21]. The doctor’s communication methods that can alleviate patient tension include the use of verbal and non-verbal behaviors [22,23]. To evaluate a doctor’s ability to communicate effectively with patients, AI patients should learn to respond appropriately to small-talk questions, not just those related to clinical diagnosis. Thus, we use small talk, a typical conversational element, as common data so that AI patients can suitably respond to the doctor’s verbal behavior. Personalized data includes the unique clinical information of each patient. Given its association with the medical field, acquiring actual medical interview data poses challenges due to the potential risk of personal information leakage. To address this problem, we introduce a synthetic data generator aimed at collecting near-real-world, standardized patient diagnosis scenarios.

The contributions of this paper are summarized as follows: (1) We proposed a Conformer-based AI patient for dental clinical diagnosis simulation. (2) We suggested a Korean synthetic data generator capable of creating Korean SP scenarios based on the given SP information. (3) We utilized the Conformer, which has been primarily used in speech processing, for Korean text and compared its performance with the Transformer, known for its outstanding performance in the existing NLP domain.

The paper is structured as follows. In the next section, we present the AI patient training data and the proposed Korean SP scenario data generator employed in this study. We also outline the deep learning model architecture and hyperparameters used for AI patient implementation, along with the performance evaluation of each model. Section 3 presents the results of the predicted answers generated by each model and provides a discussion of these outcomes. Finally, in Section 4, we draw conclusions and present future research directions.

## 2. Materials and Methods

### 2.1. Datasets for Korean Standardized AI patient

We utilized two datasets in AI Patient training. The first, common data, is employed to cultivate small-talk skills, allowing models to engage in natural conversations with people. The second is personalized data synthesized using our proposed SP scenario data generator. Personalized data pertain to information about specific standardized patients that the model aims to replicate. Descriptions of each dataset are presented in the subsequent subsections.

Figure 1 provides an overview of the data utilized for AI patient training and the scenario simulation using the generated patient. As various SP data are produced through the proposed data generator, each model learns both personalized and common data to become standardized AI patients with distinct patient characteristics. These AI patients can flexibly respond not only to clinical diagnosis questions but also to conversational exchanges aimed at easing the atmosphere.

#### 2.1.1. Common Data

During actual patient interviews, the conversation does not always focus solely on diagnosis. We trained the model on common data, including small talk, to enable it to naturally respond to casual discourse that doctors use to alleviate patient tension. Common data were collected from AI-Hub, a public data provision website for AI operated by Korean government agencies [24]. Common data encompass not only small talk but also COVID-19 counseling data from the Korea Centers for Disease Control and Prevention. Due to the nature of disease-related counseling, questions about patient information or symptoms appear, which can help the model learn responses related to general conversation and basic medical history.

#### 2.1.2. Data Generator for Standardized Patient Scenario

Owing to the nature of data associated with the medical field, collecting interview data from actual dentists proved to be challenging. Moreover, Korean is less prevalent in academia compared to English, further complicating the task of gathering relevant data. Consequently, we collected data for simulation by implementing a Korean data generator that creates the doctor’s questions and patient’s answers based on several patient diagnosis scenario scripts provided to actors. An overview of the generator is depicted in Figure 2 below.

When patient information is provided as input, the data generator produces questions and answers for each category that reflect the patient information. By inputting various patient details into the data generator, corresponding standardized patient scenarios can be created.

Our data generator consists of five main types of questions: patient identification, clinical symptoms, medical history, lifestyle, and diagnosis. Each question type has a subtype, leading to a total of 19 detailed categories. The question types and subtypes are listed in Table 1, while the content of the questions appearing in each subtype can be found in Table 2.

Our data generator created questions by applying data variation techniques to the doctor’s questions found in the diagnostic scenario script. We used a data variation approach to augment our dataset and train the model on a diverse range of syntactic representations. Data variation techniques include affix transformation, word transformation, and independent word addition. The affix transformation is a method that considers the characteristics of the Korean language. Unlike English, which is an isolating language, Korean is agglutinative, meaning one or more morphemes are attached to the root of the word to form a complete word. As a result, the meaning of sentences remains consistent, but various syntactic expressions are possible through changes in the affix alone. This approach helps generate a robust language model capable of handling different sentence expression methods. Word transformation is a method of changing words in a sentence into other similar words. This method allows the model to learn synonym expressions. Independent Word Addition is a method of adding interjections, honorifics, and conjunctive adverbs to a sentence, which are words that do not have a close relationship with other elements of the sentence. This method helps the model identify non-critical words on its own and robustly deduce responses without additional Korean preprocessing, even if speech contains unnecessary words when designing AI patients combined with an automatic speech recognition system in the future. Table 3 shows the results of applying the data variations.

### 2.2. Deep Learning for Standardized AI patient

Our AI patient is a dialogue system. Two representative methods for implementing dialogue systems include the retrieval-based model and the generative model. The retrieval-based model selects stored responses to pre-prepared queries, ensuring that there are no grammatical mistakes in inference results. However, this approach is limited as it cannot process questions without a predefined response. In contrast, the generative model can generate responses based on the learned data for unexpected questions [25]. We adopt a generative model structure to implement AI patients capable of generating responses without being restricted by input questions. The structure and hyperparameters of the generative model used are detailed in the subsequent subsections.

#### 2.2.1. Transformer

The Transformer architecture has brought significant changes to the existing RNN-based time series data processing. Although the sequential characteristics of RNN-based neural networks have the advantage of reflecting information from previous points in time, these characteristics dilute the advantages of GPU computation because they prevent parallelization of data processing. Furthermore, as the input sequence lengthens, it can lead to the vanishing gradient problem, where information is not transferred to the earlier vectors during backpropagation. Although studies into LSTM, GRU, and attention mechanisms have alleviated these problems, the sequential characteristics of RNN still remain.

The Transformer, like an RNN-based Seq2Seq model, is composed of an encoder and a decoder but does not employ RNN. To alleviate potential issues with traditional RNN-based models, the Transformer introduces techniques such as multi-head self-attention, position-wise feedforward neural networks, and positional encoding. Multi-head self-attention refers to performing self-attention operations in parallel for the specified number of heads. The self-attention mechanism obtains query (Q), key (K), and value (V) vectors from each word vector, and then uses these vectors to perform attention operations to calculate the association between each word within the input sequence. This approach performs parallel operations on the words in the sequence, unlike RNN, which performs sequential operations. As a result, it can alleviate long-term dependency issues and enable the model to capture global context information within the sentence. The position-wise feedforward neural net takes the output of the multi-head self-attention as input and applies a fully connected layer and activation function. It is similar to a standard feedforward network (FFNN) and enables parallel processing, which results in the advantage of improved computational complexity. The Transformer receives all information at once, which can cause it to disregard the order of elements within the input sequence. To address this issue, positional encoding adds information about the position of each word to its embedding vector using sinusoidal functions. This enables the model to consider the order of elements within the sequence during processing. All these techniques are crucial factors that have enabled the Transformer to outperform RNN-based models. The structure of the Transformer is shown in Figure 3, and the explanation of the Korean Word Tokens represented in the figure is presented in Section 2.2.3.

#### 2.2.2. Conformer

Thanks to self-attention, the Transformer has the ability to model global context information, but it lacks the ability to extract granular local feature patterns. On the other hand, CNN can extract local feature information such as edges or shapes of images through kernel filters, but larger models are required to capture global information. The Conformer is a structure designed to combine the local information capture function of CNN with the global information capture function of the Transformer. The Conformer encoder consists of a multi-head self-attention module, a convolution module, and a feed-forward module surrounding them. The standard Conformer encoder consists of a multi-head self-attention module, a convolution module, and a macaron-shaped feed-forward module [18]. To ensure stable training even as the model depth increases, pre-norm residual units [26] and dropout are applied to all modules of the Conformer. For example, the multi-head self-attention module in the Conformer sequentially performs computations by placing multi-head self-attention layers and dropout in the pre-norm residual units. The convolution module starts with a pointwise convolution and Gated Linear Unit (GLU) activation [27], followed by 1D convolution and batch normalization steps. Then, data features are processed through swish activation [28] and pointwise convolution. The feed-forward module is composed of two linear transformations and a non-linear activation function, similar to the feed-forward step in the Transformer model. However, it is distinguished from the Transformer by adopting swish activation as an activation function. Figure 4 illustrates the structure of the standard Conformer encoder composed of these modules.

The Conformer’s sandwich structure and half-step feed-forward layer were inspired by Macaron-Net [29], replacing the Transformer’s original feed-forward layer in this manner. The Conformer block can be mathematically expressed using the following formula:$$\begin{array}{c} h^{\prime }=x+\frac{1}{2}FFNN\left(x\right),\\ h^{\prime \prime }=h^{\prime }+MHSA\left(h^{\prime }\right),\\ h^{\prime \prime \prime }=h^{\prime \prime }+Conv\left(h^{\prime \prime }\right),\\ y=Layernorm\left(h^{\prime \prime \prime }+\frac{1}{2}FFNN\left(h^{\prime \prime \prime }\right)\right), \end{array}$$

Here, *x* represents the input of the Conformer encoder block, and *y* represents the output of the block. *FFNN* stands for feed-forward module, *MHSA* stands for multi-headed self-attention module, and *Conv* stands for convolution module.

#### 2.2.3. Korean Tokenization Text Embedding

In this paper, Korean sentences were tokenized based on morphemes, and a modified version of the Mecab tokenizer called Mecab-ko [30] was used for this purpose. Tokenization in NLP refers to dividing a given corpus into specific units called tokens. This includes dividing sentences into words or paragraphs into sentences. However, unlike English, word tokenization in the Korean corpus is different. In English, independent words can be distinguished based on white space, allowing word tokenization based on white space. However, Korean is an agglutinative language, which means that even the same word can be recognized as a different word depending on the attached morpheme. Therefore, it is not appropriate to tokenize based on white space. In Korean NLP, it is necessary to tokenize based on morphemes that separate roots and morphemes. To do this, we use Mecab-ko, an adjusted version of the Japanese morphological analysis engine Mecab [31], to create tokens based on morphemes, and perform embedding based on them.

In NLP, embedding is a crucial process that represents textual data as numerical vectors in a high-dimensional space, enabling computers to understand and process natural language data. This task is also referred to as an upstream task. In order to ensure high-quality results for downstream tasks, which are the main objectives of NLP, it is crucial to have effective upstream tasks that capture rich context information of natural language and represent it in vector space [14,32,33,34]. To achieve this, we employed the FastText algorithm [34] for text embedding in this study.

FastText is an embedding algorithm that was developed after Word2Vec [33] and shares a similar embedding mechanism. However, unlike Word2Vec, which treats words as indivisible units, FastText assumes that there are n-gram units of words within a single word (e.g., tri-grams “orange” = ora, ran, ang, nge). This approach allows FastText to infer embedding values for out-of-vocabulary or misspelled words. FastText holds a competitive edge over other word-embedding techniques that cannot extract vectors for out-of-vocabulary words.

We used the Gensim [35] FastText module to obtain embedding values for each word using the FastText algorithm. The FastText hyperparameters were configured with *vector size* = 256, *window size* = 2, *minimum count* = 1, and trained for 20 epochs using *skip-gram* and *hierarchical softmax* methods.

#### 2.2.4. Hyperparameters of Models

All deep learning models were implemented using the Transformer and Conformer modules provided by Pytorch [36] and Torchaudio [37], respectively, and the hyperparameters applied to each model are shown in Table 4. We trained the models using the cross-entropy loss function and the Adam optimizer with $learning\;rate=5\times 10^{-4}$, $\beta_{1}=0.9$, $\beta_{2}=0.98$, $eps=1\times 10^{-9}$, and early stopping was applied to prevent overfitting.

The Conformer refers to an encoder specifically designed for extracting local and global features from input data, and it does not have a clearly defined counterpart for a decoder (e.g., the Transformer consists of a Transformer encoder and Transformer decoder). To implement a Conformer-based AI patient model, a decoder is essential for translating the encoded information into language. In this paper, we constructed a Conformer-based AI patient model that utilizes a Transformer decoder as a decoder of the Conformer.

### 2.3. Performance Evaluation

In order to evaluate the model’s performance, the Bilingual Evaluation Understudy (BLEU) score and Word Error Rate (WER) were adopted as evaluation metrics. The BLEU score evaluates how much the sentence predicted by the model matches the correct sentence [38]. The BLEU score is calculated using the following equation:$$\begin{array}{c} MP_{n}\;=\;\frac{\sum_{g\in h}min\left(C\left(g,h\right),C\left(g,r\right)\right)}{\sum_{g\in h}C\left(g,h\right)},\\ BP=\left\{\begin{array}{c} 1,\;if\;h>r\\ e^{1-\;\frac{h}{r}},\;if\;h\leq r \end{array}\right.\\ BLEU\;=\;BP\times exp\left(\overset{N}{\underset{n=1}{\sum }}w_{n}logMP_{n}\right), \end{array}$$

Here, $MP_{n}$ represents modified n-gram precision, and C represents the number of times the word g in n-gram contained in a given sentence appears. h means the predicted sentence, and r means the correct answer sentence. BP is a brevity penalty, which prevents situations where a high BLEU score is obtained when the h is shorter than r. N is the maximum length of the n-gram. $w_{n}$ means the weight applied to each n-gram, and in this experiment, a weight of 0.25 was applied to all n-grams.

WER evaluates word-level errors between real and predicted sentences. WER is calculated as follows:$$WER\;=\;\frac{I+D+S}{N},$$

Here, I represents the number of words incorrectly added to the predicted sentence, and D means the number of words that do not appear in the predicted sentence. S is the number of words substituted between the correct sentence and the predicted sentence, and N is the total number of words in the correct sentence.

The AI patients were trained using both common data and personalized data. The common data comprise 67,124 pairs of question-and-answer data collected from AI-Hub, while the personalized data for one standardized patient created through the data generator consist of 18,378 pairs. The training dataset contains 78,150 data pairs, which consist of 67,124 data pairs from the common data and 11,026 data pairs accounting for 60% of the total personalized data. Due to the primary objective of the AI patient model being the accurate prediction of answers to diagnostic questions, both the validation and test datasets were comprised exclusively of personalized data. Each of these datasets contains a total of 3676 data pairs, which correspond to exactly half of the remaining personalized data.

We analyzed the impact of model structures on performance with a focus on the Conformer and Transformer. Figure 5 shows the loss and performance of each model in the validation dataset. In Figure 5, the blue color represents the Transformer model, and the red color represents the Conformer model. The left plot demonstrates that the Conformer has a lower loss value and converges faster than the Transformer. This trend is also observable in the right plot. In the right plot, the BLEU score is represented by a solid line, while the WER is denoted by a dotted line. The Conformer’s BLEU score quickly surpassed 90% before 10 epochs and maintained the convergence value, whereas the Transformer exhibited a gradual upward trend up to 60 epochs and ended prematurely. Similarly, for WER, the Conformer showed a distinct downward trend, while the Transformer stalled with a relatively gentle downward trend. The quantitative performance of each model is presented in Table 5. In the test dataset, the Transformer’s results were 2.58% lower in BLEU score and 2.94% higher in WER compared to the results in the validation dataset, achieving 92.21% and 8.08%, respectively. The Conformer also exhibited a slight decrease in performance in the test dataset compared to the validation dataset, but ultimately achieved a BLEU score of 96.13% and a WER of 1.34%. This demonstrates a performance improvement of 3.92% and 6.74% over the Transformer, respectively.

## 3. Results and Discussion

The performance gap between the Transformer and Conformer models might be attributed to the model size. To verify this hypothesis, we examined whether performance improvement would be observed when the parameter size of the Transformer was equal to that of the Conformer. Table 6 provides a concise summary of the number of layers and parameters implemented in the Transformer model, as well as the corresponding performance outcomes. The experiment revealed that within the conditions of the utilized dataset, as the Transformer’s parameters approached those of the Conformer, performance degradation occurred in the validation dataset due to overfitting on the training dataset.

Table 7 presents the answers generated by each model for the questions in the test dataset. The top two rows of the Transformer model’s results inferred answers that reflect the provided SP information, suggesting that the Transformer model can make accurate predictions for questions related to personalized data that are distinct from common data. In contrast, rows three to six display responses that do not comply with the given scenario. This result suggests that when the model encounters personalized data similar to common data, the inference outcome tends to be biased towards the common data. As an example, the Transformer-based model often generates responses such as “yes,” which frequently appear in common data. This indicates that although the Transformer model may possess basic communication capability, it lacks the proficiency needed to understand the context required for engaging in natural dialogues with humans.

In the case of the Conformer, examining the first three rows from the top reveals that, unlike the Transformer, the Conformer can infer answers that align with the patient’s role and comprehend the intent of the questions. However, the Conformer model also provided out-of-context answers for certain question types. Rows four to six of the Conformer results in Table 7 pertain to questions about the SP’s pain location. While the fourth row demonstrates comprehension of the question’s intent and provides an appropriate answer, the fifth and sixth rows present unrelated answers despite being the same question type. Both the Conformer and Transformer models exhibited bias due to the abundance of data; however, the Conformer showed a more robust performance compared to the Transformer. This finding confirms that the model architecture, which incorporates local and global information, plays a crucial role in enhancing learning efficiency. Through this experiment, we confirmed that the Conformer structure demonstrates superior performance not only in speech data but also in text format, surpassing the Transformer. Moreover, we verified that the Conformer-based AI patient can play a more natural patient role than the Transformer-based model when utilized in clinical practice.

Although the Conformer-based AI patient also exhibited challenges in comprehending the intent of certain types of questions, we believe that if more systematic data collection is conducted for clinical diagnosis, these issues can be effectively addressed. Furthermore, this approach holds potential to serve as an educational system for professional occupations not only in dentistry but also in other fields.

## 4. Conclusions

This study proposed a Conformer-based AI patient and Korean SP scenario data generator for dental clinical diagnosis simulation. The proposed data generator can generate questions for doctors to understand a patient’s condition and responses that reflect the SP’s information. Additionally, through a data variation approach, it can create a variety of syntactic expressions. The utilization of this data generator has the advantage of enabling the creation of standardized patients for clinical training and collecting real clinical diagnostic question and answer data without any cost or time constraints. For AI patient training, common data containing responses to natural conversations and personalized data, including clinical information specific to the designated patient role, were used. To confirm the learning efficiency of the Conformer structure based on these data, we performed a performance analysis comparing it to the Transformer, which had a significant impact on NLP, using BLEU score and WER as evaluation metrics. As a result of our experiment, it was confirmed that the Conformer outperformed the Transformer in generating improved responses. The use of AI patients in SP simulation has the potential to enhance the limited clinical diagnosis practice environment faced by medical professionals due to environmental, cost, and recruitment challenges. Moreover, this approach could be extended to other fields of medicine and nursing through additional data collection and refinement. Our study demonstrates the implementation of standardized patients by integrating deep learning with dental clinical education. We hope that this approach can lead to significant transformations in medical education. In future research, we intend to incorporate ASR, TTS, and VR systems to develop a simulated clinical diagnostic system that closely mirrors real-world environments.

## Figures and Tables

**Figure 1 bioengineering-10-00615-f001:**
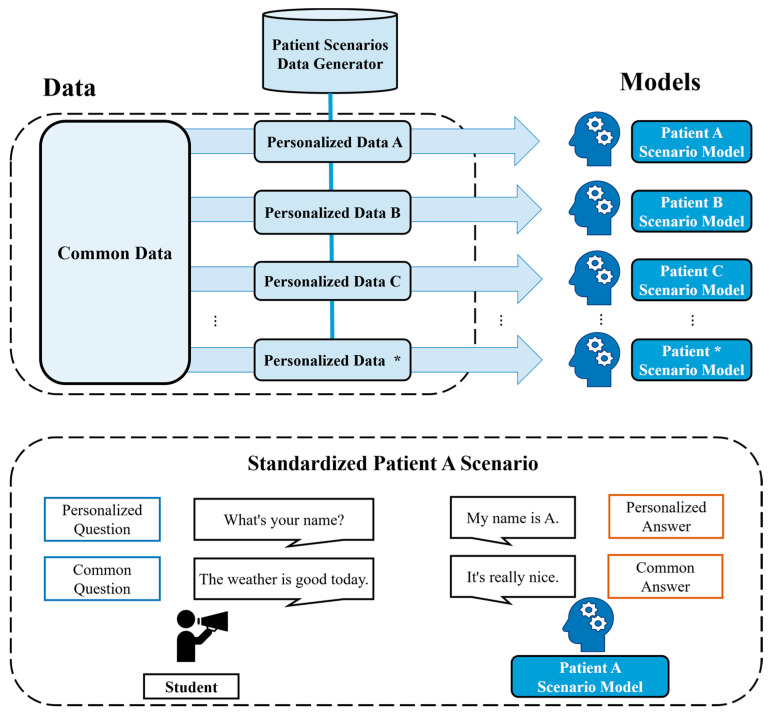
Overview of the standardized AI patient clinical diagnosis simulation. The symbol (*) in the figure represents the number of SP models to be generated.

**Figure 2 bioengineering-10-00615-f002:**
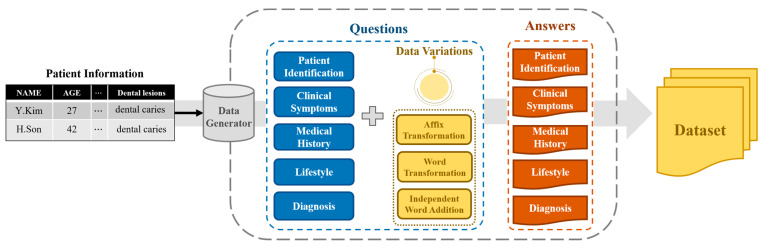
Overview of the standardized patient scenario data generator.

**Figure 3 bioengineering-10-00615-f003:**
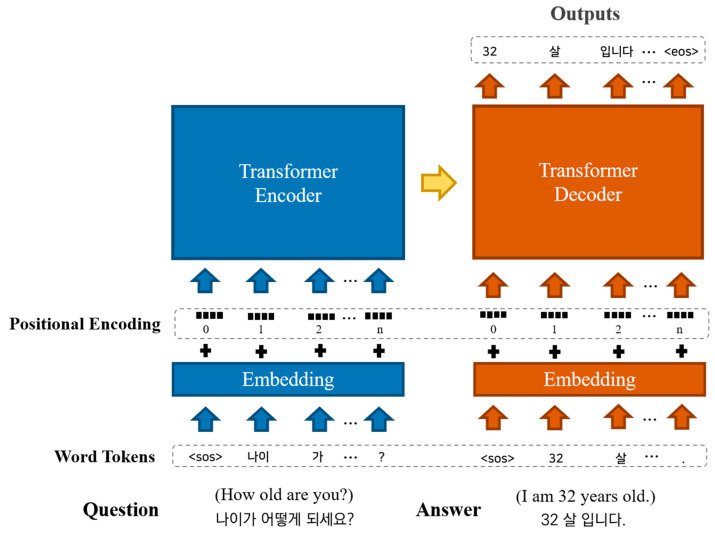
Structure of the Transformer model.

**Figure 4 bioengineering-10-00615-f004:**
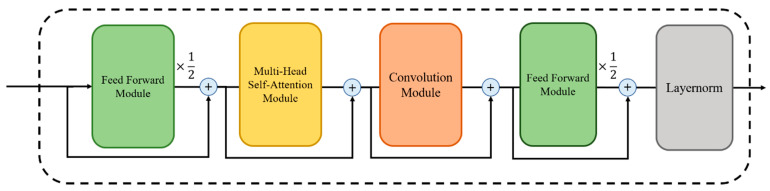
The structure of the Conformer encoder.

**Figure 5 bioengineering-10-00615-f005:**
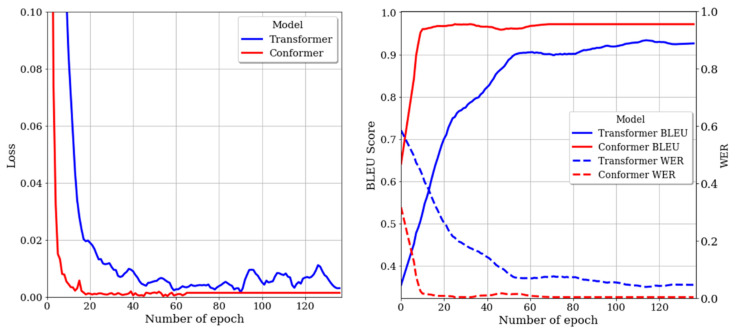
(**Left**) Losses of the model on the validation dataset. (**Right**) BLEU score and WER of the models obtained from the validation dataset.

**Table 1 bioengineering-10-00615-t001:** The question types and subtypes of the proposed data generator.

Question Types	Subtypes
Patient Identification	Name Identification
	Age Identification
Clinical Symptoms	Sore Spot
	Time of Onset
	Pain-inducing Factors
	Pain Duration
	The Intensity of Pain
Medical History	Medical History Listening
	Medications Check
	Dental History Listening
	Experience of Anesthesia
Lifestyle	Smoking Status
	Alcohol Consumption Check
Diagnosis	Visual Inspection
	Percussion Test
	Mobility Test
	Electrical Pulp Test
	Pulp Thermal Test
	Radiographic Examination

**Table 2 bioengineering-10-00615-t002:** Representative examples of questions for each subtype are shown. The blank space that appears for age is filled with the age information of the input patient.

Question Types	Language	Examples
Name Identification	English	What’s your name?
	Korean	성함이 어떻게 되세요?
Age Identification	English	Are you ○○ years old?
	Korean	○○ 세 맞으시죠?
Sore Spot	English	Which tooth hurts?
	Korean	어디 치아가 아프세요?
Time of Onset	English	When did this symptom start?
	Korean	증상이 언제부터 시작됐나요?
Pain-inducing Factors	English	When does your tooth hurt?
	Korean	어떨 때 치아가 아프시나요?
Pain Duration	English	How long did the pain last?
	Korean	통증이 얼마 동안 지속되던가요?
The Intensity of Pain	English	How much do you think the pain is out of 10?
	Korean	통증이 10점 만점으로 어느 정도 되는 것 같으세요?
Medical History Listening	English	Are you being treated at any other hospital besides the dentist?
	Korean	치과 말고 다른 병원에서 치료받고 계신 게 있나요?
Medications Check	English	Do you take any medicine regularly?
	Korean	주기적으로 드시는 약 있으세요?
Dental History Listening	English	What kind of dental treatment have you had?
	Korean	어떤 치과 치료받아보셨나요?
Experience of Anesthesia	English	Have you ever been anesthetized during dental treatment?
	Korean	치과 치료할 때 마취하신 적 있으세요?
Smoking Status	English	Do you smoke?
	Korean	담배 피우시나요?
Alcohol Consumption Check	English	Do you drink often?
	Korean	음주를 자주 하시나요?
Visual Inspection	English	I’ll check it out myself.
	Korean	직접 확인해 보도록 하겠습니다.
Percussion Test	English	I’ll pat the tooth on the side that hurts.
	Korean	아픈 쪽 치아를 두드려 보겠습니다.
Mobility Test	English	I’ll shake your sore teeth.
	Korean	아프신 치아를 흔들어 보겠습니다.
Electrical Pulp Test	English	Let’s do a dental nerve test using electricity.
	Korean	전기를 이용해서 치아 신경 검사를 해보겠습니다.
Pulp Thermal Test	English	We will examine the dental nerve response according to temperature.
	Korean	온도에 따른 치아 신경 반응을 검사하겠습니다.
Radiographic Examination	English	I’ll take a radiograph to check for inflammation.
	Korean	염증 확인을 위해 방사선 사진을 찍어 보겠습니다.

**Table 3 bioengineering-10-00615-t003:** The data variation types and application examples. Colored text indicates changes due to the application of variations.

Data Variations		Language	Examples
Affix Transformation	Original	English	What’s your name?
		Korean	성함이 어떻게 되세요?
	Variation	English	What is your name?
		Korean	성함이 어떻게 되시나요?
Word Transformation	Original	English	When did the pain start?
		Korean	언제부터 통증이 있었나요?
	Variation	English	When did you start feeling symptoms?
		Korean	언제부터 증상이 느껴졌나요?
Independent Word Addition	Original	English	How much alcohol do you usually drink?
		Korean	평소에 술은 얼마나 드세요?
	Variation	English	Sir, how much alcohol do you usually drink?
		Korean	선생님, 평소에 술은 얼마나 드세요?

**Table 4 bioengineering-10-00615-t004:** Hyperparameters to consider for each model.

Hyperparameters	Transformer	Conformer
Word Embedding	FastText	FastText
Embedding Size	(256)	(256)
Hidden Size	(256)	(256)
Number of Encoder Layers	(5)	(4)
Number of Decoder Layers	(5)	(4)
Multi-Head	(8)	(8)
FFNN Size	(512)	(512)
Dropout	(0.4)	(0.4)
Batch Size	(256)	(256)
Learning Rate	$\left(5\times 10^{-4}\right)\;$	$\left(5\times 10^{-4}\right)$
Depthwise Convolution Kernel Size	-	(31)
Number of Params	12.3 M	13.6 M
Epoch	137	65

**Table 5 bioengineering-10-00615-t005:** Quantitative performance results obtained using the validation and test datasets.

Model	Validation		Test	
	BLEU (%)	WER (%)	BLEU (%)	WER (%)
Transformer	94.79	5.14	92.21	8.08
Conformer	96.40	1.04	96.13	1.34

**Table 6 bioengineering-10-00615-t006:** Performance results from increasing the parameters of the Transformer model.

Number of Layers	Number of Params	Validation	
		BLEU (%)	WER (%)
[5, 5]	12.3 M	94.79	5.14
[6, 6]	13.6 M	87.38	9.71
[7, 7]	14.9 M	52.82	46.92

**Table 7 bioengineering-10-00615-t007:** The answer prediction results for the Transformer and Conformer models on the test dataset.

Model	Questions	Answers	Predicted Answers
Transformer	Sir, then I will diagnose you from now on.	Yes, thank you.	Oh, yes I see.
	Sir, on a scale of 10 for pain, how much pain did you get?	It’s like 7 points.	It’s like 7 points.
	Patient, do you ever feel discomfort in your teeth?	It hurts when food gets stuck.	I’ve been sick since a week ago.
	Hello, What is your name?	My name is Taeju Han.	Yes.
	Hello, when did your symptoms first start?	I’ve been sick since a week ago.	Today.
	Patient, how long did the pain last?	It’s like the pain disappears when the food is gone.	I don’t think there is much of anything like that.
Conformer	Hello, what is your name?	My name is Taeju Han.	My name is Taeju Han.
	Hello, when did your symptoms first start?	I’ve been sick since a week ago.	I’ve been sick since a week ago.
	Patient, how long did the pain last?	It’s like the pain disappears when the food is gone.	I don’t think there is much of anything like that.
	Patient, which tooth is bothering you that made you come here today?	My lower right molar hurts.	The lower right molar is in pain.
	Which part of the tooth is inconvenient?	My lower right molar hurts.	Yes, I am in Jeju Island.
	What tooth do you have pain in?	My lower right molar hurts.	Yes, I see.

## Data Availability

Not applicable.

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
