# Peer review of "Conformer-Based Dental AI Patient Clinical Diagnosis Simulation Using Korean Synthetic Data Generator for Multiple Standardized Patient Scenarios"

_bioengineering, 2023, doi:10.3390/bioengineering10050615_

Round 1

Reviewer 1 Report

The authors propose a Conformer-based dental AI-patient to replace actors in clinical diagnosis simulation and a Korean SP scenario data generator for its training.The results look encouraging and motivating. But there are still some contents, which need be revised in order to meet the requirements of publish. A number of concerns listed as follows:

(1)   The abstract should be improved. Your point is your own work that should be further highlighted.

(2)   In the introduction, the authors should clearly indicate the contributions and innovations of this paper.

(3)

(4) In Table 4 of Hyperparameters to consider for each model. How to  determine thses values? Please give reason.

(5) In order to highlight the introduction, some latest references should be added to the paper for improving the reviews part and the connection with the literature. https://doi.org/10.3389/fendo.2022.1057089; http://dx.doi.org/10.1016/j.oceaneng.2022.113424; http://dx.doi.org/10.1109/TCSS.2022.3152091; https://doi.org/10.1016/j.ins.2023.03.142 and so on.

(6) There are some grammatical errors seen in the paper. Check carefully for a few clerical errors and formatting issues.

There are some grammatical errors seen in the paper. Check carefully for a few clerical errors and formatting issues.

Author Response

 We would like to express our gratitude to Reviewer 3 for the constructive comments and valuable suggestions that have led us to improve our work significantly. Please see below for the specifics in response to the reviewer’s comments.

Comments 1)

(1)   The abstract should be improved. Your point is your own work that should be further highlighted.

Response 1)

We really appreciate this comment. We have revised the abstract as follows:

Before)

(Abstract)

The goal of clinical practice education is to develop the ability to apply theoretical knowledge in a clinical setting and to foster growth as a professional healthcare provider. One effective meth-od of achieving this is through the utilization of Standardized Patients (SP) in education, which familiarizes students with real patient interviews and allows educators to assess their clinical performance skills. However, the use of SP techniques is limited by high costs and a shortage of professional educators available to train actors. In this study, we pro-pose a Conformer-based dental AI-patient to replace actors in clinical diagnosis simulation and a Korean SP scenario data generator for its training. Our Korean SP scenario data generator is de-vised to generate SP scenarios based on the provided patient information, using pre-prepared questions and answers. In the AI-Patient training process, two types of data are employed: common data and personalized data. The common data is employed to develop natural general conversation skills, while personalized data, which is the SP scenario, is utilized to learn specific clinical information relevant to a patient's role. Based on these data, to evaluate the learning efficiency of the Conformer structure, a comparison was conducted with the Transformer using the BLEU score and WER as evaluation metrics. Experimental results showed that the Conformer-based model demonstrated a 3.92% and 6.74% improvement in BLEU and WER performance compared to the Transformer-based model, respectively. The dental AI-patient for SP simulation presented in this paper has the potential to be applied to other medical and nursing fields, pro-vided that additional data collection processes are conducted.

After)

(Abstract)

The goal of clinical practice education is to develop the ability to apply theoretical knowledge in a clinical setting and to foster growth as a professional healthcare provider. One effective meth-od of achieving this is through the utilization of Standardized Patients (SP) in education, which familiarizes students with real patient interviews and allows educators to assess their clinical performance skills. However, SP education faces challenges such as the cost of hiring actors and the shortage of professional educators to train them. In this paper, we aim to alleviate these issues by utilizing deep learning models to replace the actors. We employ the Conformer model for the implementation of the AI patient, and we develop a Korean SP scenario data generator to collect data for training responses to diagnostic questions.Our Korean SP scenario data generator is devised to generate SP scenarios based on the provided patient information, using pre-prepared questions and answers. In the AI-Patient training process, two types of data are employed: common data and personalized data. The common data is employed to develop natural general conversation skills, while personalized data, which is the SP scenario, is utilized to learn specific clinical information relevant to a patient's role. Based on these data, to evaluate the learning efficiency of the Conformer structure, a comparison was conducted with the Transformer using the BLEU score and WER as evaluation metrics. Experimental results showed that the Conformer-based model demonstrated a 3.92% and 6.74% improvement in BLEU and WER performance compared to the Transformer-based model, respectively. The dental AI-patient for SP simulation presented in this paper has the potential to be applied to other medical and nursing fields, pro-vided that additional data collection processes are conducted.

Comments 2)

(2)   In the introduction, the authors should clearly indicate the contributions and innovations of this paper.

Response 2)

We really appreciate this comment. We have clearly revised our contributions as follows:

Before)

(Introduction)

In this paper, we propose a Conformer-based AI-patient for dental clinical diagnosis simulation and a Korean SP synthetic data generator for its training, focusing on dentistry and NLP in Korean. Additionally, we analyze the performance improvement effects of the model structure using quantitative evaluation metrics.

After)

(Introduction)

The contribution of this paper is summarized as follows:

  1. We proposed a Conformer-based AI-patient for dental clinical diagnosis simulation
  2. We suggested a Korean synthetic data generator capable of creating Korean SP scenarios based on the given SP information.
  3. We utilized the Conformer, which has been primarily used in speech processing, for Korean text and compared its performance with the Transformer, known for its outstanding performance in the existing NLP domain.

Comments 3)

(3)

Response 3)

Due to the content for item 3 being left blank, we were unable to verify the relevant requirements.

Comments 4)

In Table 4 of Hyperparameters to consider for each model. How to determine thses values? Please give reason.

Response 4)

The determining of these values is heavily influenced by the original paper, as well as by the size of the data we utilize. Despite the small number of dataset we use, increasing the size of the model may lead to overfitting, resulting in a decrease in the model's generalization ability. The parameter values shown in Table 4 can be understood as the results of the parameters that we derived through our empirical experiments while avoiding overfitting.

Comments 5)

(5)    In order to highlight the introduction, some latest references should be added to the paper for improving the reviews part and the connection with the literature. https://doi.org/10.3389/fendo.2022.1057089; http://dx.doi.org/10.1016/j.oceaneng.2022.113424; http://dx.doi.org/10.1109/TCSS.2022.3152091;

https://doi.org/10.1016/j.ins.2023.03.142 and so on.

Response 5)

We really appreciate this comment. To highlight the Introduction, we have made revisions by referring to the reference as follows:

Before)

(Introduction)

…However, utilizing actors as SPs for practice comes with the requirement of expensive labor costs, dedicated manpower, and significant time for actor training [4]. As a solution to these problems, in this study, we aim to replace actors with deep learning.

After)

… However, utilizing actors as SPs for practice comes with the requirement of expensive labor costs, dedicated manpower, and significant time for actor training [4]. As deep learning has advanced, there have been reports of research utilizing it for disease diagnosis [5], but applications for clinical diagnosis simulation are rare. In this study, we aim to utilize deep learning to alleviate the reliance on actors in the SP method.

[5]: Li, M.; Zhang, J.; Song, J.; Li, Z.; Lu, S. A Clinical-Oriented Non-Severe Depression Diagnosis Method Based on Cognitive Behavior of Emotional Conflict. IEEE Transactions on Computational Social Systems 2022.

Comments 6)

(6) There are some grammatical errors seen in the paper. Check carefully for a few clerical errors and formatting issues.

Response 6)

We really appreciate this comment. We have identified and corrected grammatical errors as much as could be found in the manuscript.

Before)

(2.3 Performance Evaluation)

…Common data and personalized data were used to train AI-patients. The common data consists of 67,124 pairs of question-and-answer data collected from AI-Hub, and the personalized data for one standardized patient created through the data generator consists of 18,378 pairs. The training dataset included 78,150 common data items and 11,026 data items, which represent 60% of personalized data. As the model should focus on predicting responses to diagnostic questions, the validation and test datasets each contained 3,676 data pairs, representing 20% of the remaining personalized data…

…In the left plot, it is evident that the Conformer has a lower loss value and converges faster than the Transformer…

(4. Conclusions)

The proposed data generator can produce anticipated questions for doctors to understand a patient's condition and responses that reflect the SP's information.

After)

(2.3 Performance Evaluation)

…The AI-patients were trained using both common data and personalized data. The common data comprises 67,124 pairs of question-and-answer data collected from AI-Hub, while the personalized data for one standardized patient created through the data generator consists of 18,378 pairs. The training dataset contains 78,150 data pairs, which consist of 67,124 data pairs from the common data and 11,026 data pairs accounting for 60% of the total personalized data…

…In the left plot, it demonstrates that the Conformer has a lower loss value and converges faster than the Transformer…

(4. Conclusions)

The proposed data generator can generate questions for doctors to understand a patient's condition and responses that reflect the SP's information.

We really appreciated the constructive comments and valuable suggestions again.

Reviewer 2 Report

It's a borderline paper - its overal recommendation is somewhere between "Reconsider after major revision" and "Reject". The main problem is the low quality of presentation, which practically disallows to repeat and re-use the experiments performed by the Authors.

For example the architecture presented in Fig. 3 ("Structure of the Transformer model") contains elements like "Positional vector" and "Word tokens", which are not described in a way which enables my own construction of similar system. Also "Figure 4. The structure of the Conformer encoder" contains, for example the module called "multi-head self-attention module". How does it work? Its description in the part of the text between lines 218 and 223 does not give me a chance to develop my own module of that type.

In the first lines of section "2.2.3. Text embedding" the Authors write that "processes that represent text as numbers in vector space are essential" and do not describe how they did it (only mention the Word2Vec algorithm, probably described in detail in pos. [28]).

There are also many other doubts and inconsistencies which justify my opinion expressed above.

The manuscript should be checked by a native speaker, as there are grammar errors (too many to list them individually).

Author Response

 We would like to express our gratitude to Reviewer 1 for the constructive comments and valuable suggestions that have led us to improve our work significantly. Please see below for the specifics in response to the reviewer’s comments.

Comment 1-1)

For example the architecture presented in Fig. 3 ("Structure of the Transformer model") contains elements like "Positional vector" and "Word tokens", which are not described in a way which enables my own construction of similar system.

Response 1-1)

We apologize for the confusion caused by the lack of specific explanation of the terms "Position vector" and "Word tokens" in Fig. 3 ("Structure of the Transformer model") of the manuscript.

We have modified Fig. 3 by changing "Positional vector" to "Positional Encoding" in order to avoid confusion. Additionally, we have provided a reference to the original paper and a brief explanation, as our Positional Encoding follows the same approach as described in the paper.

(Added)

(2.2.1 Transformer)

The positional encoding used in this paper uses the same sinusoidal function as presented in [12] ...

Before)

(2.2.1 Transformer)

The Transformer is composed of an encoder and decoder, just like the Seq2Seq model. However, the Transformer uses a Feed-Forward Neural Network (FFNN) instead of the RNN-based model of the existing structure to enable parallel computation and introduces positional encoding to reflect the order information of words lost in this process….

Figure 3. Structure of the Transformer model.

After)

(2.2.1 Transformer)

The Transformer is composed of an encoder and decoder, just like the Seq2Seq model. However, the Transformer uses a Feed-Forward Neural Network (FFNN) instead of the RNN-based model of the existing structure to enable parallel computation and introduces positional encoding to reflect the order information of words lost in this process. The positional encoding used in this paper uses the same sinusoidal function as presented in [12] ...

Figure 3. Structure of the Transformer model.

Comment 1-2)

"Word tokens", which are not described in a way which enables my own construction of similar system.

Response 1-2)

We really appreciate this comment. We really agree with your comment that there is insufficient explanation about how the "Word tokens" appearing in Fig. 3 were tokenized and how it can be reproduced.

We have made revisions in Section 2.2.3 by modifying the title to "Korean Tokenization and Text Embedding" and including an explanation and relevant references.

(Added)

, and the explanation of the Korean Word Tokens represented in the figure is presented in Section 2.2.3.

Before)

(2.2.1. Transformer)

… The structure of the Transformer is shown in Figure 3.

(2.2.3. Text embedding)

In NLP, embedding processes that represent text as numbers in vector space are essential so that computers can understand text; this is called an upstream task…

After)

(2.2.1. Transformer)

The structure of the Transformer is shown in Figure 3, and the explanation of the Korean Word Tokens represented in the figure is presented in Section 2.2.3.

(2.2.3. Korean Tokenization and Text Embedding)

In this paper, Korean sentences are tokenized based on morphemes, and a modified version of the Mecab tokenizer called Mecab-ko, is used for this purpose [29]. Tokenization in NLP refers to dividing a given corpus into specific units called tokens. This includes dividing sentences into words or paragraphs into sentences. However, unlike English, the word-tokenization in Korean corpus is different. In English, independent words can be distinguished based on white space, allowing word tokenization based on white space. However, Korean is an agglutinative language, which means that even the same word can be recognized as a different word depending on the attached morpheme. Therefore, it is not appropriate to tokenize based on white space. In Korean NLP, it is necessary to tokenize based on morphemes that separate roots and morphemes. To do this, we use Mecab-ko, an adjusted version of the Japanese morphological analysis engine Mecab [30], to create tokens based on morphemes, and perform embedding based on them.

In NLP, embedding is a crucial process that represents textual data as numerical vectors in a high-dimensional space, enabling computers to understand and process natural language data. This task is also referred to as an upstream task. In order to ensure high quality results for downstream tasks, which are the main objectives of NLP, it is crucial to have effective upstream tasks that capture rich context information of natural language and represent it in vector space [13, 31, 32, 33]. To achieve this, we employ the FastText algorithm [33] for text embedding in this study.

Comment 2)

Also "Figure 4. The structure of the Conformer encoder" contains, for example the module called "multi-head self-attention module". How does it work? Its description in the part of the text between lines 218 and 223 does not give me a chance to develop my own module of that type.

Response 2)

We really appreciate this comment. We really agree that an explanation of how each module in the Conformer encoder is structured is necessary. Therefore, we have added a reference to the paper consulted for module design and explanations of the structure of each module.

Before)

(2.2.2. Conformer)

…The Conformer encoder consists of a multi-head self-attention module, a convolution module, and a macaron-shaped feed-forward module [17]. Figure 4 depicts the overall structure of the Conformer encoder.

After)

(2.2.2. Conformer)

…The standard Conformer encoder consists of a multi-head self-attention module, a convolution module, and a macaron-shaped feed-forward module [17]. To ensure stable training even as the model depth increases, pre-norm residual units [25] and dropout are applied to all modules of the Conformer. For example, the multi-head self-attention module in the Conformer sequentially performs computations by placing multi-head self-attention layers and dropout in the pre-norm residual units. The convolution module starts with a pointwise convolution and Gated Linear Unit (GLU) activation [26], followed by 1D convolution and batch normalization steps. Then, data features are processed through swish activation [27] and pointwise convolution. The feed-forward module is composed of two linear transformations and a non-linear activation function, similar to the feed-forward step in the Transformer model. However, it is distinguished from the Transformer by adopting swish activation as an activation function. Figure 4 illustrates the structure of the standard Conformer encoder composed of these modules.

Comment 3)

In the first lines of section "2.2.3. Text embedding" the Authors write that "processes that represent text as numbers in vector space are essential" and do not describe how they did it (only mention the Word2Vec algorithm, probably described in detail in pos. [28]).

Response 3)

We apologize for the confusion caused by the lack of clear explanation on how we performed FastText. To make it clear, we have added an explanation of the method we used.

Additionally, we have added an information in Section 2.2.4 Hyperparameters of Models regarding the loss function used during model training and the hyperparameter values provided to the optimizer.

Added)

(2.2.3. Korean Tokenization and Text Embedding)

We use Gensim [34] FastText module to obtain embedding values for each word using the FastText algorithm. The FastText hyperparameters are configured with vector size = 256, window size = 2, minimum count = 1, and it train for 20 epochs using skip-gram and hierarchical softmax methods.

(2.2.4. Hyperparameters of Models)

Table 4 lists the hyperparameters to be considered for each model. We train the models using cross-entropy loss function and Adam optimizer with learning rate = ,  = 0.9,  = 0.98, and eps = , and early stopping is applied to prevent overfitting…

Before)

(2.2.3. Text embedding)

FastText is an embedding algorithm that emerged after Word2Vec and uses a similar embedding mechanism as Word2Vec [32]. However, while Word2Vec considers words as indivisible units, FastText assumes that there are n-gram units of words within a single word (e.g., tri-grams "orange" = ora, ran, ang, nge).

(2.2.4. Hyperparameters of Models)

Table 4 lists the hyperparameters to be considered for each model. The Adam optimizer was used as the optimization function for the models, and early stopping was ap-plied to prevent overfitting…

After)

(2.2.3. Korean Tokenization and Text Embedding)

FastText is an embedding algorithm that was developed after Word2Vec [32] and shares a similar embedding mechanism. However, unlike Word2Vec which treats words as indivisible units, FastText assumes that there are n-gram units of words within a single word (e.g., tri-grams "orange" = ora, ran, ang, nge)…

We use Gensim [34] FastText module to obtain embedding values for each word using the FastText algorithm. The FastText hyperparameters are configured with vector size = 256, window size = 2, minimum count = 1, and it train for 20 epochs using skip-gram and hierarchical softmax methods.

(2.2.4. Hyperparameters of Models)

Table 4 lists the hyperparameters to be considered for each model. We train the models using cross-entropy loss function and Adam optimizer with learning rate = ,  = 0.9,  = 0.98, and eps = , and early stopping is applied to prevent overfitting…

We really appreciated the constructive comments and valuable suggestions again.

Reviewer 3 Report

This paper discusses an attempt to create standardized patients using generative models. It is an extremely ambitious and intriguing paper, and the use of generative models is quite original and worthy of presentation. However, there are a few areas that could be improved:

1.Please clarify the model's loss function.

2.If possible, provide a demo using Hugging Face; otherwise, make the source code (even for a pre-trained model) publicly available on platforms such as GitHub.

3.The description of the dataset split from lines 297 to 303 is confusing. For example, it says, "Common data and personalized data were used to train AI-patients. The common  data consists of 67,124 pairs of question-and-answer data collected from AI-Hub, and the personalized data for one standardized patient created through the data generator consists of 18,378 pairs. The training dataset included 78,150 common data items and 11,026 300 data items, which represent 60% of personalized data." However, this would result in the total dataset containing fewer data items than the training dataset. Since data partitioning is a critical aspect, please provide a clear and accurate description.

Author Response

We would like to express our gratitude to Reviewer 2 for the constructive comments and valuable suggestions that have led us to improve our work significantly. Please see below for the specifics in response to the reviewer’s comments.

Comments 1)

  1. Please clarify the model's loss function.

Response 1)

We really appreciate this comment. We have revised the manuscript to include the loss function in addition to the optimizer parameters in Section 2.2.4 Hyperparameters of Models as follows.

Added)

We train the models using cross-entropy loss function and Adam optimizer with learning rate = ,  = 0.9,  = 0.98, and eps = , and early stopping is applied to prevent overfitting…

Before)

(2.2.4. Hyperparameters of Models)

Table 4 lists the hyperparameters to be considered for each model. The Adam optimizer was used as the optimization function for the models, and early stopping was ap-plied to prevent overfitting…

After)

(2.2.4. Hyperparameters of Models)

Table 4 lists the hyperparameters to be considered for each model. We train the models using cross-entropy loss function and Adam optimizer with learning rate = ,  = 0.9,  = 0.98, and eps = , and early stopping is applied to prevent overfitting…

Comments 2)

  1. If possible, provide a demo using Hugging Face; otherwise, make the source code (even for a pre-trained model) publicly available on platforms such as GitHub.

Response 2)

We really appreciate this comment. We really agree that uploading a demo is crucial to prove the authenticity of the paper. However, unfortunately, we have not obtained the necessary approval from the institution associated with this project, making it difficult for us to release the code and model.

Comments 3)

  1. The description of the dataset split from lines 297 to 303 is confusing. For example, it says, "Common data and personalized data were used to train AI-patients. The common data consists of 67,124 pairs of question-and-answer data collected from AI-Hub, and the personalized data for one standardized patient created through the data generator consists of 18,378 pairs. The training dataset included 78,150 common data items and 11,026 300 data items, which represent 60% of personalized data." However, this would result in the total dataset containing fewer data items than the training dataset. Since data partitioning is a critical aspect, please provide a clear and accurate description.

Response 3)

We are sorry that this part is not clear in the manuscript. We appreciate your feedback and have revised clarify the content.

Before)

(2.3 Performance Evaluation)

…Common data and personalized data were used to train AI-patients. The common data consists of 67,124 pairs of question-and-answer data collected from AI-Hub, and the personalized data for one standardized patient created through the data generator consists of 18,378 pairs. The training dataset included 78,150 common data items and 11,026 data items, which represent 60% of personalized data. As the model should focus on predicting responses to diagnostic questions, the validation and test datasets each contained 3,676 data pairs, representing 20% of the remaining personalized data…

After)

(2.3 Performance Evaluation)

…The AI-patients were trained using both common data and personalized data. The common data comprises 67,124 pairs of question-and-answer data collected from AI-Hub, while the personalized data for one standardized patient created through the data generator consists of 18,378 pairs. The training dataset contains 78,150 data pairs, which consist of 67,124 data pairs from the common data and 11,026 data pairs accounting for 60% of the total personalized data…

We really appreciated the constructive comments and valuable suggestions again.

Round 2

Reviewer 1 Report

This paper can be accepted now.

Author Response

We really appreciate for your acceptance

Reviewer 2 Report

I appreciate the fact that the Authors tried to take into account the reviewer's comments and expanded some parts of the manuscript. However, I still believe that the evaluated presentation of the solution to the problem does not allow the implementation of the approach (described in a manuscript) to solve a similar class of issues. The architecture and principle of operation of the key modules of the system presented in the manuscript are described roughly, with numerous references to the literature.

The authors have not clearly listed their own original elements or modifications of known methods, developed to implement the system presented in the manuscript. All the above justifies my opinion about the signifance of content and scientific soundness of the reviewed manuscript.

Author Response

We would like to express our gratitude to Reviewer 1 for the constructive comments and valuable suggestions that have led us to improve our work significantly. Please see below for the specifics in response to the reviewer’s comments.

Comment 1)

The architecture and principle of operation of the key modules of the system presented in the manuscript are described roughly, with numerous references to the literature.

Response 1)

We really appreciate this comment. We have revised the manuscript by providing additional explanations on the operations performed by each block of the Transformer and the benefits it provides.

Before)

(2.2.1. Transformer)

The Transformer is composed of an encoder and decoder, just like the Seq2Seq model. However, the Transformer uses a Feed-Forward Neural Network (FFNN) instead of the RNN-based model of the existing structure to enable parallel computation and introduces positional encoding to reflect the order information of words lost in this process. The posi-tional encoding used in this paper uses the same sinusoidal function as presented in [13]. In addition, the Transformer adopts self-attention, which can calculate correspondence regardless of the distance between words in the input sequence using query, key, and val-ue vectors, demonstrating remarkable efficiency in alleviating long-term dependency problems. Self-attention uses multi-head to reflect various semantic characteristics of words. Multi-head self-attention is a method of expressing the characteristics of a word by calculating self-attention for each head and concatenating all values. All these techniques are crucial factors that have enable the Transformer to outperform RNN-based models in terms of performance. The structure of the Transformer is shown in Figure 3, and the explanation of the Korean Word Tokens represented in the figure is presented in Section 2.2.3.

After)

(2.2.1. Transformer)

The Transformer, like an RNN-based Seq2Seq model, is composed of an encoder and a decoder but does not employ RNN. To alleviate potential issues with traditional RNN-based models, the Transformer introduces techniques such as multi-head self-attention, position-wise feedforward neural networks, and positional encoding. The multi-head self-attention refers to performing self-attention operations in parallel for the specified number of heads. The self-attention mechanism obtains query (Q), key (K), and value (V) vectors from each word vector, and then uses these vectors to perform attention operations to calculate the association between each word within the input sequence. This approach performs parallel operations on the words in the sequence, unlike RNN which perform sequential operations. As a result, it can alleviate long-term dependency issues and enable the model to capture global context information within the sentence. The position-wise feedforward neural net takes the output of the multi-head self-attention as input and applies a fully-connected layer and activation function. It is similar to a standard feedforward network (FFNN) and enables parallel processing, which results in the advantage of improved computational complexity. The Transformer receives in all information at once, which can cause it to disregard the order of elements within the input sequence. To address this issue, positional encoding adds information about the position of each word to its embedding vector using sinusoidal functions. This enables the model to consider the order of elements within the sequence during processing. All these techniques are crucial factors that have enable the Transformer to outperform RNN-based models. The structure of the Transformer is shown in Figure 3, and the explanation of the Korean Word Tokens represented in the figure is presented in Section 2.2.3.

Comment 2)

The authors have not clearly listed their own original elements or modifications of known methods, developed to implement the system presented in the manuscript.

Response 2)

We apologize for any lack of clarity in our manuscript. In response to this comment, we have added explanations about the tools used in implementing the system. This information will help facilitate the reimplementation of the model presented in the manuscript.

Before)

(2.2.4. Hyperparameters of Models)

Table 4 lists the hyperparameters to be considered for each model. We train the models using cross-entropy loss function and the Adam optimizer with , , , , and early stopping is applied to prevent overfitting. The Conformer-based AI-patients have only changed the encoder of the Transformer structure to the Conformer encoder. Since the Conformer is specialized in extracting local and global features within the input data, it is appropriate to change only the encoder part.

After)

(2.2.4. Hyperparameters of Models)

All deep learning models are implemented using the Transformer and Conformer modules provided by Pytorch [36] and Torchaudio [37] libraries, respectively, and the hyperparameters applied to each model are shown in Table 4. We train the models using cross-entropy loss function and the Adam optimizer with , , , , and early stopping is applied to prevent overfitting.

The Conformer refers to an encoder specifically designed for extracting local and global features from input data, and it does not have a clearly defined counterpart for a decoder (e.g., the Transformer consists of a Transformer encoder and Transformer decoder). To implement a Conformer-based AI patient model, a decoder is essential for translating the encoded information into language. In this paper, we construct a Conformer-based AI patient model that utilizes a Transformer decoder as a decoder of the Conformer.

We really appreciated the constructive comments and valuable suggestions again.
